# TAKING ROCKET ON AN EFFICIENCY MISSION: A DISTRIBUTED SOLUTION FOR FAST AND ACCURATE MULTIVARIATE TIME SERIES CLASSIFICATION

## ABSTRACT

Nowadays, with the rising number of sensors in sectors such as healthcare and industry, the problem of multivariate time series classification (MTSC) is getting increasingly relevant and is a prime target for machine and deep learning solutions. Their expanding adoption in real-world environments is causing a shift in focus from the pursuit of ever higher prediction accuracy with complex models towards practical, deployable solutions that balance accuracy and parameters such as prediction speed. An MTSC solution that has attracted attention recently is ROCKET, based on random convolutional kernels, both because of its very fast training process and its state-of-the-art accuracy. However, the large number of features it utilizes may be detrimental to inference time. Examining its theoretical background and limitations enables us to address potential drawbacks and present LightWaveS: a distributed solution for accurate MTSC, which is fast both during training and inference. Specifically, utilizing a wavelet scattering transformation of the time series and distributed feature selection, we manage to create a solution which employs just 2,5% of the ROCKET features, while achieving accuracy comparable to recent deep learning solutions. LightWaveS also scales well with more nodes and large numbers of channels. In addition, it can significantly reduce the input size and also provide insight to an MTSC problem by keeping only the most useful channels. We present three versions of our algorithm and their results on training time, accuracy, inference speedup and scalability. We show that we achieve speedup ranging from 9x to 65x compared to ROCKET during inference on an edge device, on datasets with comparable accuracy.

## 1 INTRODUCTION

Time series classification is the task of characterizing a series of values observed in sequential moments in time as belonging to one of two or more categories, or classes. There has been extensive work on univariate time series classification with machine and deep learning methods, as observed in surveys such as Bagnall et al. (2017). In practice, as noted in Ruiz et al. (2021), many problems are described by more than one channels of information, turning the task into multivariate time series classification (MTSC). Some recent contributing factors have been the development of smaller and cheaper sensors for the measurement of various quantities, the general advancement of Internet of Things and the engagement with inherently more complex problems in sectors such as healthcare and industry, which benefit from additional information channels. In addition, the rise of edge computing and deployment of models in such environments makes it necessary to also consider the conditions and resources of the computational nodes on which those models will execute. This change in perspective makes it important to include, apart from accuracy, variables such as prediction speed or throughput in the evaluation of a model's suitability for a given task.

In the recent evaluation of advances in MTSC (Ruiz et al., 2021), one solution that shows good performance is ROCKET (Dempster et al., 2019), both in terms of accuracy and training time. ROCKET utilizes random convolutions to transform the time series channels and then extracts two features per convolution that are used with a linear classifier. The evolution of ROCKET, MINIROCKET (Dempster et al., 2020), is proposed as the new default variant of ROCKET by its authors and utilizes only one feature per kernel (percentage of positive values), thereby halving

the features. It also utilizes other optimizations to speed up ROCKET in general, and uses a minimally random approach with a given set of kernels. Another variant, MultiRocket (Tan et al., 2021), uses more features per convolution to achieve better accuracy at the expense of transformation time. Both ROCKET and MINIROCKET have been developed for univariate time series. Although the authors of MINIROCKET characterize their provided repository code for multivariate time series as naive, it is the same algorithmic logic in the form of a ROCKET extension in the sktime (Löning et al., 2019) repository that achieves the impressive performance mentioned above.

In (MINI)ROCKET, the authors encourage research into more sophisticated approaches for multivariate time series. We can start by identifying potential points that can be improved. One point with (MINI)ROCKET that we try to address in our work is that the default number of features, although beneficial to accuracy, can be highly redundant, and extreme for some datasets, leading to unnecessarily high transformation and inference times during deployment. In addition, due to the fixed process and number of generated features, there is a large discrepancy in the representation (features per channel) across datasets with different dimensions. Moreover, the indiscriminate inclusion of all channels in the feature generation is susceptible to uninformative series. We present experiments to illustrate the above points in section A.1. Finally, the stochastic element, especially the random channel combination, although beneficial to accuracy, does not offer significant interpretability.

Based on this analysis, we propose LightWaveS, a framework for fast, distributed transformation of multivariate time series based on convolutional kernels, wavelet scattering and feature selection, for lightweight and accurate classification with linear classifiers. Our solution aims to keep the successful aspects of the ROCKET model family, such as the short transformation time, the arbitrary convolutional kernel approach and the few descriptive features per kernel. On top of that, LightWaveS adds well-studied signal theory, multi-node[1] distribution and smart feature selection to address the drawbacks identified above. The trade-off that we propose compared to (MINI)ROCKET is clear: we take advantage of additional computational resources during training time to keep the short duration and also save time during inference. Our contributions with LightWaveS are:

- We introduce the concept of wavelet scattering based on arbitrary kernels
- We achieve accuracy comparable to state-of-the-art in the majority of the UEA datasets (Bagnall et al., 2018), using only a fraction of the number of features used by (MINI)ROCKET
- We distribute the framework to achieve training time comparable to (MINI)ROCKET on 1 node, and shorter on multiple nodes
- We achieve linear scaling of training time with the number of channels, tested in experiments with up to 900 channels
- Depending on the dataset size, we achieve good speedup with additional nodes, tested in experiments with up to 8 nodes
- We achieve inference speedups ranging from 9x to 65x compared to ROCKET on an edge device in datasets with comparable accuracy

With this work, we take an efficiency-centric approach, focusing on the practicality and the inference speed of a model that may be deployed on an edge device, without necessarily trying to surpass the state-of-the-art in terms of accuracy.

## 2 RELATED WORK

### 2.1 MULTIVARIATE TIME SERIES CLASSIFICATION

Due to the recent rise in popularity of the deep learning field, there is a multitude of models that can be easily adapted to incorporate the additional dimension of MTSC (Ismail Fawaz et al., 2019). A detailed evaluation of recent MTSC methods appears in Ruiz et al. (2021). Convolutions in particular, either 1-D or with more dimensions, are a very popular module when dealing with time series, as seen in models such as TapNet (Zhang et al., 2020), InceptionTime (Fawaz et al., 2020) and OS-CNN (Tang et al., 2021) among others. In these works, convolutions are combined in

---

[1]With the term *node* we refer to a computing node in a distributed (networked) system.

various architectures with other modules, such as fully connected networks, and achieve impressive classification accuracy. The majority of those deep learning models takes a significant amount of time and memory to train even on GPU nodes, ranging from hours to even days, depending on the dataset size (Ruiz et al., 2021). In contrast, our method takes less than 20 minutes to transform, train and test all 30 UEA datasets on a single CPU node with a linear classifier. If we distribute the solution on e.g. 8 CPU nodes, this time drops to under 4 minutes. An additional challenge is the interpretability of deep learning models. The usage of wavelets by LightWaveS and their method of application gives the potential to interpret the result based on signal theory and the input channel filtering helps to extract useful conclusions about a given time series problem.

## 2.2 FEATURE EXTRACTION AND SELECTION

Deep learning models implicitly extract features from the input along their first layers. In contrast, explicitly extracting features to use with classifiers has also been visited in multiple ways. We have already described how the *ROCKET family extracts features from the convolutions of random kernels with the input, an idea that has also been explored before, such as in Farahmand et al. (2017). Another example is tsfresh (Christ et al., 2018), which extracts a large number of predefined statistical features from the time series and then through feature selection reduces them to the most useful. Similarly, catch22 (Lubba et al., 2019) is a solution that uses only 22 predefined characteristics to transform time series, aiming for a very fast transformation. A different approach is presented in WEASEL+MUSE (Schäfer & Leser, 2018), which extracts features based on a bag-of-patterns approach and selects the most useful based on a $\chi^2$ test. LightWaveS, similarly to MINIROCKET, depends on an arbitrary set of convolutions and only 4 statistical features, extracted however from the coefficients of wavelet scattering. Due to the speed of the convolution operation and the simplicity of the features, we can achieve extremely fast training and inference times, while giving the model enough complexity to accommodate difficult datasets, where predefined statistical features on the raw time series values may not be descriptive enough.

## 2.3 DISTRIBUTED TRAINING

As the size of the deep learning models and the amount of input data increase, it is increasingly difficult to perform training on a single node. For that reason, distribution of the training process across multiple nodes is becoming increasingly necessary (Mayer & Jacobsen, 2020). This distribution can be implemented with methods such as data parallelism, where each node applies the same operations on different parts of the input data, with frequent communication among the nodes to update the model. Distribution can also be applied on solutions that do not depend on deep learning, such as tsfresh mentioned above, which supports operation on a cluster through Dask (Dask Development Team, 2016). LightWaveS is distributed using MPI in a data-parallel way, but in contrast to the communication heavy training of DL models, there is efficient and minimal communication between the worker nodes and the central coordinator. Specifically, the nodes only send a limited number of feature scores and descriptions once during the execution, making communication a trivial percentage of the whole process.

## 2.4 WAVELETS

Wavelets are localized waveforms (as seen in Figure 1 on the right) and are a well-studied method in signal processing that has been used extensively in the analysis of time series (Mallat, 1999). There is vast literature with methods and applications of wavelets on all types of problems, ranging from healthcare to audio analysis, and well-studied and developed families of wavelets suitable for specific applications (Addison, 2017; Merry & Steinbuch, 2005). There are also numerous approaches that combine wavelets with machine and deep learning methods, either as implicit or explicit feature extractors (Li et al., 2021; Wang et al., 2018). A seminal work is Bruna & Mallat (2013), where the concept of a wavelet scattering network using a Morlet wavelet is introduced, in combination with linear and support vector machine classifiers for hand-written digit classification and texture recognition. This method was constructed to be invariant to translations of the input and stable to small deformations. It has also been recently used in combination with deep learning networks for specific applications (Soro & Lee, 2019; Jin & Duan, 2020).

LightWaveS aims to combine the strong points of these works under a single generalized framework, with a focus on efficiency. We aim to bridge the gap between ROCKET and the wavelet theory, and we progress to the next logical step of wavelet scattering. We keep this approach lightweight, both in depth and paths of the scattering, so we can apply it to time series channels on a massive scale in a very short time. The arbitrary base set of wavelets can potentially be extended based on expert opinion, backed by the solid theory behind wavelets and their applications, making LightWaveS a suitable platform for experimentation on solutions for MTSC problems. Finally, the hierarchical feature filtering leads to the most relevant output features of the scattering coefficients being selected.

# 3 PROPOSED FRAMEWORK

## 3.1 PRELIMINARIES

### 3.1.1 MINIROCKET FUNDAMENTALS

(MINI)ROCKET is primarily based on the convolution operation, in which a kernel $\boldsymbol{k}$ with size $l$ ($\boldsymbol{k} \in \mathbb{R}^l$), bias $\beta$ and dilation factor $d$ is used to calculate a sliding dot product with a 1-D input $\boldsymbol{x} \in \mathbb{R}^L$ of size $L$ and produce an output $\boldsymbol{x}'$, where each element is calculated as (Dempster et al., 2019):

$$\boldsymbol{x}'_i = \sum_{j=0}^{l-1} \boldsymbol{k}_j \cdot \boldsymbol{x}_{i+d*j} + \beta \tag{1}$$

MINIROCKET generates a large number (10000 by default) of those kernels. The weights are selected from an empirically chosen subset of 84 kernels of length 9 with weights in $\{2, -1\}$. This has been chosen to limit the possible weight combinations but is not unique for the purpose, and other lengths, different values or weights drawn from $\sim \mathcal{N}(0, 1)$ are equally effective (Dempster et al., 2020). The only thing that is important is that the kernel weights have sum 0. The bias is drawn from the convolution output, and the dilation is drawn from the set $\mathbb{D} = \{\lfloor 2^e \rfloor\}$, where e$\sim \mathcal{U}(0, m)$, with $m$ such that the length of the dilated kernel does not exceed that of the input. Finally, half of the kernels use padding, which appends zeros to either side of the input so that $\boldsymbol{x}'$ and $\boldsymbol{x}$ have the same dimension. The feature extracted from each of the 10000 outputs is the percentage of positive values, and is used for the final classification.

### 3.1.2 WAVELET SCATTERING

Wavelet scattering is the process of applying wavelet transforms in a cascading manner (Bruna & Mallat, 2013), combined with non-linearities and pooling. The wavelet transform is a method used to approximate a signal using a set of wavelets which originate from a "mother" wavelet $\Psi(t)$, scaled by $s$ and shifted by $b$ (Young, 2012). Each such wavelet can be described as:

$$\Psi_{s,b}(t) = \frac{1}{\sqrt{s}} \Psi(\frac{t-b}{s}) \tag{2}$$

Intuitively, we can relate these wavelets to the convolution filters that we discussed above, with the dilation being the scale of the wavelets (how "narrow" or "wide" they are), and the shift parameter $b$ corresponding to the starting point of the convolution on the input ($i+d*j$). This connection between convolutional networks and the scattering architecture has also been thoroughly explored in Mallat (2016). We can see an example of a wavelet convolution in Figure 1, where the response is strong at the points where the sliding wavelet "matches" the signal. In addition, the wavelet set created from a mother wavelet is parallel to the way that MINIROCKET has a fixed set of kernel weights, for which different paddings and dilations are randomly selected, generating the child kernels.

In the wavelet scattering transform, on each level $\lambda$, the previous result is convoluted with each wavelet $\psi_{\lambda_n}$ (kernel) and a complex modulus operator is applied before propagating the result to the next level, such that:

$$U[\lambda] = |U[\lambda - 1] * \psi_{\lambda_n}| \tag{3}$$

and the scattering coefficients that result from each level are

$$S[\lambda] = U[\lambda] * \phi \tag{4}$$

where $\phi$ is an averaging kernel. Since on every level of the scattering transform there can be multiple candidate wavelets (kernels), there is a geometric progression of potential paths, as can be seen in the graphic representation of the process in Figure 1.

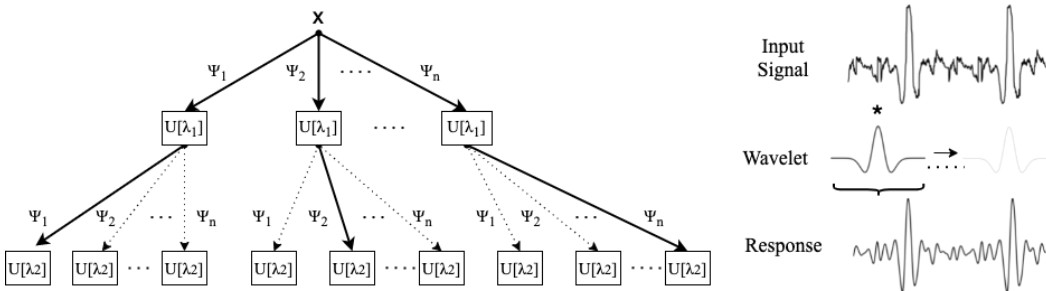

Figure 1: 2-level wavelet scattering (Bruna & Mallat, 2013) and wavelet convolution example

Throughout the above intuitive description, we connected the concepts of MINIROCKET with the wavelet theory. In MINIROCKET the prototype kernels have zero mean, which corresponds to a desirable constraint of mother wavelets (Young, 2012). Under this new context, we can re-consider MINIROCKET as a classifier based on convolutions with random child wavelets of an arbitrary set of mother wavelets. In addition, although not being wavelet scattering, MINIROCKET can be intuitively mapped on the first level, with its kernel set equivalent to the first level set, $\{\Psi_{0,1,s}, ..., \Psi_{0,n,s}\}$, with $s$ drawn from $\mathbb{D}$ as we said above. Since each kernel is slid across the input, we can think of $b$ as taking all discrete values from 0 to $l_{input} - l_{kernel} * s$ for each candidate wavelet, and the convolution output (response) as being the combined response of all shifted wavelets, as portrayed in Figure 1. However, MINIROCKET extracts the features before the application of any modulus operator, so it does not satisfy the rest of the wavelet scattering transform requirements.

## 3.2 ALGORITHM

Based on the above observations, we improve the approach and reach the crux of LightWaveS: Lightweight Wavelet Scattering based on random wavelets. The term lightweight refers both to the system optimizations of the framework that make it fast, such as the distribution, as well as the fact that only a reduced set of wavelet scattering paths are computed. Although it is established that not all paths need to be considered in a scattering network (Bruna & Mallat, 2013), since we are aiming for fast training and inference, we take this notion to its limit. We compute all coefficients for the given kernels and dilations for the first level, but we consider only one path per first level output for the second level. In this way, we limit significantly the memory and computation time required for the extracted features, while accepting the trade-off of losing some descriptive coefficients.

The choice of the kernels to be applied in the second layer is not immediately clear. We can consider heuristics such as selecting kernels whose first level coefficients gave features with high correlation to the classification task. However, after experimenting with such heuristics, and even with up to four second layer paths, we found that the final classification results were not better than the simple approach of applying the same kernel and dilation, a concept intuitively and loosely similar to the process of a discrete wavelet transform (Sundararajan, 2016). Thus, we end up computing only the paths shown in bold in Figure 1, which means that the same kernel is applied twice consecutively. In addition, we limit the depth to two layers, as a good balance point between computational speed and informative coefficients, as observed in Bruna & Mallat (2013).

The main steps of LightWaveS are the following: Initially, the dataset is split among the nodes across the channel dimension. For larger datasets, the nodes receive only a sample of the total training examples, in order to speed up computation. This sample selection is the only source of randomness in the algorithm. Then, on each node, the kernels are generated, which for our purpose are based on the same subset of 84 kernels that MINIROCKET uses. We limit randomness even more, so each kernel is dilated with all dilations in the set $\{2^0, 2^1, ..., 2^5\}$. We also completely

remove bias, and the padding is common to all kernels so that the output dimension is equal to the input.

All those kernels are applied to each of the input channels according to the wavelet scattering process described above. The generated kernels have no complex part, so the non-linearity modulus operator is equal to the absolute value of the convolution output. We also downsample the $U[\lambda_1]$ result by a factor of 2 before propagating it, while keeping the same scale for the kernel, so it operates in lower frequencies. This is according to the insight in Bruna & Mallat (2013) that the most useful paths are frequency decreasing. As far as the pooling is concerned, since we want to limit the number of coefficients that will be used as features, we use max and min pooling, instead of the averaging kernel. In this way, although we accept the loss of more information, we have fewer features and we also avoid two additional convolutions per path. The features that are extracted from each scattering path are 8: the first four are the max and min value of $U[\lambda_1]$ and $U[\lambda_2]$. We selected those as the most straightforward non-linearities that can be quickly calculated during the convolution process. The other four are the percentage of positive values (ppv) and normalized longer sequence of positive values (ls) in $U[\lambda_1]$ and $U[\lambda_2]$ before the application of the modulus operator. We keep these features based on MINIROCKET and MultiROCKET respectively, since they have proven to be useful during classification.

After the feature extraction, the first selection phase is performed on each node in a supervised way using ANOVA. The main node then gathers the top scoring features and performs the final feature selection, using the minimum-redundancy, maximum-relevance algorithm (Peng et al., 2005), incrementally selecting the feature with the highest score. This score is determined by its F-statistic (relevance to the class), divided by its average correlation to the previously selected features (redundancy). In our case, we use the Pearson correlation coefficient as the correlation indicator. In both phases the feature selection methods are filter based, which are fast, classifier-independent and can be implemented to scale well with the number of features (Li et al., 2017). The design choices that are not guided or restricted by theory, such as the feature scoring function, are straightforward and focused on computational efficiency. We also performed empirical experimentation on a development subset, although without exhaustive parameter tuning of the method.

## 4 EXPERIMENTS

### 4.1 DATASETS

We select as benchmark the UEA collection of multivariate datasets (Bagnall et al., 2018), excluding *InsectWingbeat*, since due to its large size it presented issues when training ROCKET. The datasets are described in detail in Ruiz et al. (2021). Following the example of Dempster et al. (2019), we selected 15 of the 30 UEA datasets to work on when developing the method, in order to draw generic conclusions and avoid overfitting the whole collection. In addition, we prepare and use five machinery related datasets:

- **MAFAULDA**, from Machinery Fault Database (MAF) is a dataset with 8 sensor measurements on a machine fault simulator, taken under normal conditions and five different fault types. We downsample the measurements so that the input length is 1000 steps and we split the dataset in train and test with ratio 85-15 %.

- **TURBOFAN** (Saxena & Goebel, 2008), is an engine degradation simulation dataset collection, containing 4 datasets with operation simulations of engines that run until failure under different conditions, with measurements from 26 sensors. The goal is to predict the remaining useful life (RUL) of the engines. In order to turn the problem into binary classification, we prepare the dataset and the labels in a suitable way with the goal being classifying RUL as more or less than 20 operational cycles.

### 4.2 EXPERIMENTAL SETUP

All training experiments were run on nodes which have dual 8-core 2.4 GHz (Intel Haswell E5-2630-v3) CPUs and 64GB of RAM. The inference experiments are executed on a Jetson Xavier board which has an 8-core ARM CPU. We ran (MINI)ROCKET on the 29 UEA datasets and the 5 additional datasets using the default number of features (20 and 10 thousand respectively). As for

LightWaveS, we present three variants of the model, termed L1,L2 and L1L2. These versions refer to keeping the features only from the scattering level 1, level 2 or both, although we consider the L1L2 version as the default. We select 500 features as the default variant, which has shown good balance between training time, inference speed and accuracy in the development set. We use a Ridge regression classifier from the Scikit-learn package (Pedregosa et al., 2011) for all methods. We use a critical difference diagram to present the results, a popular method of comparing classifier performance across multiple datasets (Ruiz et al., 2021; Ismail Fawaz et al., 2019). This shows the ranking of the classifiers and groups the ones that do not show statistical difference, based on pairwise comparisons using a Wilcoxon signed-rank test (Wilcoxon, 1992) with Holm's alpha correction (Holm, 1979; Garcia & Herrera, 2008). The grouped classifiers appear on the diagram connected by a thick horizontal line.

## 4.3 ACCURACY RESULTS

We present the performance of LightWaveS in terms of accuracy in comparison with (MINI)ROCKET, as well as other recent solutions, namely (M)OSS-CNN (Tang et al., 2021), WEASEL+MUSE (Schäfer & Leser, 2018), TapNet (Zhang et al., 2020) and Catch22 (Lubba et al., 2019). Apart from LightWaveS and (MINI)ROCKET, the rest of the accuracy metrics are taken from the repositories of Tang et al. (2021)[2] and Dhariyal et al. (2020)[3]. Due to missing values in those metrics, we manage to compare the methods on 28 of the 30 UEA datasets.

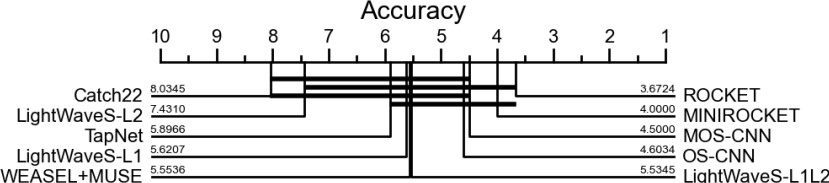

Figure 2: Mean rank of LightWaveS methods vs recent classifiers in terms of accuracy

In Figure 2 we can see the results. Although lower in rank, LightWaveS belongs in the same statistical group with (MINI)ROCKET, which have the best accuracy. In addition, it is among the ranks of more complex DL methods, such as TapNet and MOS-CNN. Out of the three variants, LightWaveS-L2 performs the worst, without the benefit of faster execution that L1 has. We can focus on (MINI)ROCKET for a more detailed comparison, and also include the 5 additional datasets. Since the aim of LightWaveS is to approach their state-of-the-art accuracy with fewer features, not necessarily surpass it, we place the LightWaveS results for all datasets in four accuracy bins compared to (MINI)ROCKET : one for higher or equal accuracy, and three bins for lower accuracy, with difference less than 0.05, between 0.05 and 0.1 and more than 0.1 respectively. In Figure 3 we see the amount of datasets in each bin. For the majority of the datasets the accuracy stays in the first 3 categories for all variants apart from L2, and the datasets with large accuracy deviation are few. In addition, we can see that just increasing the number of features to 1500 leads to improvement of the comparison results, especially in the case of MINIROCKET, showing the potential of the method to successfully tackle the harder datasets as well.

## 4.4 CHANNEL AND NODE SCALING OF TRAINING TIME

As a point of reference, ROCKET processes the whole UEA set (apart from *InsectWingbeat*) on a single node in approximately 9 minutes, MINIROCKET in 5 and LightWaveS (L1L2) in 14. (MINI)ROCKET has complexity $\mathcal{O}(kernels \cdot samples \cdot tslength)$. Since LightWaveS uses one more convolution per kernel it adds a fixed factor of two to the complexity, essentially keeping linear scalability with these variables. Thus, we can focus on the scalability of the LightWaveS training time with the additional channel dimension and with the number of distribution nodes.

---

[2] https://github.com/Wensi-Tang/OS-CNN
[3] https://github.com/mlgig/mtsc_benchmark/

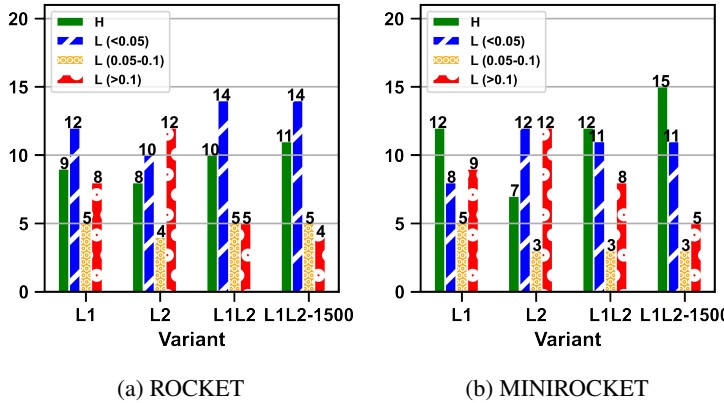

(a) ROCKET                  (b) MINIROCKET

Figure 3: Distribution of datasets based on average accuracy difference between LightWaveS variants and (MINI)ROCKET

We select *PEMS-SF* from UEA which has 963 channels, and we use subsets of the total dataset to train LightWaveS on 2 nodes, starting from 100 channels and incrementing by 100. We see in Figure 4a that all variants show (sub)linear scaling with the number of channels, which is expected since each additional channel results in a fixed amount of additional convolutions, and there are no combinations or permutations of channels. For the node scaling, we also select *FaceDetection*, with 144 channels and we measure the training time across 1,2,4 and 8 nodes. As we see in Figure 4b, with PEMS-SF the gained training speedup is more close to ideal, since there are many channels for each node to work on and the distributed computation constitutes the majority of the execution time. In contrast to that, with FaceDetection that has fewer channels, the computational cost starts being dominated by the operations on the main node (application of kernels, feature selection, final transformation), so the gained speedup falls more quickly.

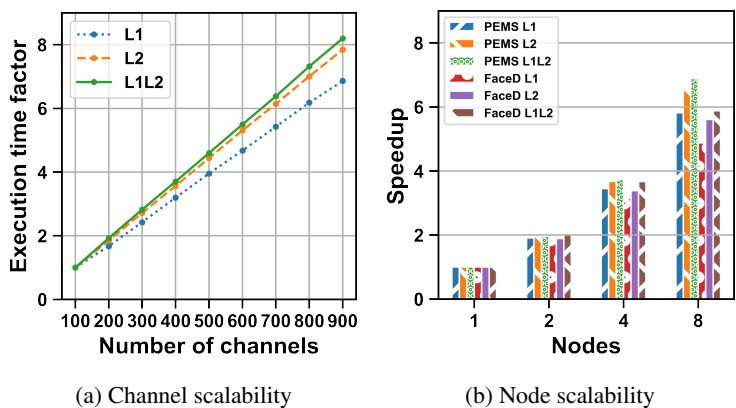

(a) Channel scalability             (b) Node scalability

Figure 4: Scalability of LightWaveS traing time with the number of (a) channels and (b) nodes

## 4.5 INFERENCE RESULTS

We manage to achieve significant speedup during inference due to the small amount of features, which heavily reduces the amount of convolutions required to transform a test sample. We perform a qualitative analysis, focusing on measuring the inference on the machinery datasets, since they could correspond to a scenario of edge deployment of a model for e.g. fault detection of a machine. However, our conclusions hold for all datasets. We can see in Figure 5 the relation between the accuracy difference of the LightWaveS variants with (MINI)ROCKET and the speedup achieved. This figure frames clearly the trade-off that we propose: it is up to the end user to select their preferred balance of speedup and accuracy, in the cases where no ideal case is available. The MF

dataset in the MINIROCKET comparison acts as an outlier in the graph due to its large length in combination with the optimizations of MINIROCKET. Our method of applying the kernels matches ROCKET's, but could also benefit from these optimizations, restoring the discrepancy.

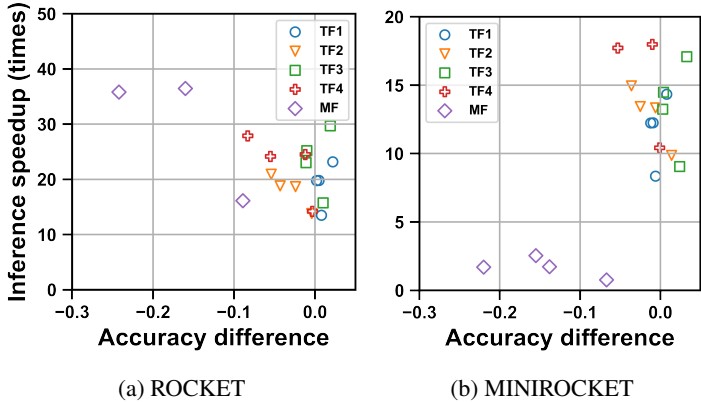

(a) ROCKET          (b) MINIROCKET

Figure 5: Average inference speedup-accuracy difference scatterplot for the 5 machinery datasets, for each of the LightWaveS variants (note the different y axis scale)

## 5   DISCUSSIONS AND CONCLUSION

Summarizing, LightWaveS offers comparable to state-of-the-art accuracy on the datasets tested, with training time similar to (MINI)ROCKET and inference time 9x-65x faster on an edge device in the case of ROCKET and 1.3x - 14x for MINIROCKET. Apart from the advantages we have mentioned, a significant advantage of LightWaveS is that it is based on wavelet theory, which has a strong theoretical base. A promising future direction is to explore expert tuning of the framework, by preparing and including in the base set well tested wavelets, with different properties such as padding or dilations depending on the use case. Another benefit is that due to the feature selection, LightWaveS can filter the channels to a subset of the originals. As seen in Table 5, this reduction ranged from 0 to 92%, with an average of 15% across all datasets and the larger datasets benefiting more. This has multiple advantages: it can give insights into which channels contain useful information for the problem, leading to knowledge extraction. On edge devices, where resources are valuable, it can free up incoming signal channels. Finally, LightWaveS can act as an initial fast channel filtering method that precedes another deep learning solution, reducing the training data required. LightWaveS can also benefit from, and is indeed orthogonal to, recent works on feature selection, since additional descriptive features can be used on the scattering coefficients, improving accuracy. Future work can include an initial, more informed selection of the wavelets, so that the wavelet scattering network can extend to more paths with wavelet combinations, as well as explicit, informed combinations of different channels.

### REPRODUCIBILITY STATEMENT

The code of LightWaveS, as well as the code used to train (MINI)ROCKET, run the inference experiments and preprocess the machinery datasets is provided as supplementary material. All accuracy and time experiments were repeated multiple times (30-100), with different seeds and the average of metrics was taken for a fairer representation of the results. The detailed metrics are provided in the Appendix. All datasets used are publicly available. There has been an effort to explicitly set the random seed for all methods that accept such a parameter, in order to enable reproducibility of the results as closely as possible.

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

# A APPENDIX

## A.1 (MINI)ROCKET ANALYSIS

Using the *sktime* repository version of the code, we can see in Figure 6a that even keeping the top half features of MINIROCKET using an unsophisticated chi-squared test does not result in a statistically significant difference in the classifier performance (rather, it slightly increases the rank). As for the channel inclusion, as an extreme case, we can add noise channels to the UEA datasets (10% of the original channel number) and we see that both solutions are affected negatively when trained on them, belonging in a statistically different group.

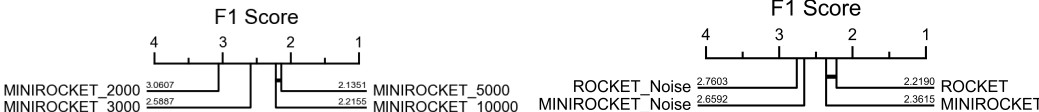

(a) Selection of fractions of the default number of features

(b) Training on normal and noisy versions of the UEA dataset

Figure 6: Mean ranks of (MINI)ROCKET under different experiments, in terms of F1 score

## A.2 DETAILED EXPERIMENT METRICS

Below are the detailed metrics on accuracy, inference speedup and channel reduction.

Table 1: Detailed accuracy metrics of LightWaveS-L1L2 and (MINI)ROCKET

| Dataset | L1L2 | | ROCKET | | MINIROCKET | |
|---|---|---|---|---|---|---|
| | Mean | Std | Mean | Std | Mean | Std |
| ArticularyWordRecognition | 0.997 | 0.000 | 0.993 | 0.000 | 0.993 | 0.002 |
| AtrialFibrillation | 0.267 | 0.000 | 0.067 | 0.000 | 0.124 | 0.023 |
| BasicMotions | 1.000 | 0.000 | 1.000 | 0.000 | 1.000 | 0.000 |
| CharacterTrajectories | 0.983 | 0.000 | 0.992 | 0.001 | 0.992 | 0.001 |
| Cricket | 0.931 | 0.000 | 1.000 | 0.000 | 0.986 | 0.000 |
| DuckDuckGeese | 0.440 | 0.000 | 0.505 | 0.033 | 0.696 | 0.027 |
| ERing | 0.959 | 0.000 | 0.986 | 0.003 | 0.981 | 0.003 |
| EigenWorms | 0.962 | 0.000 | 0.901 | 0.008 | 0.954 | 0.007 |
| Epilepsy | 0.978 | 0.000 | 0.988 | 0.004 | 1.000 | 0.000 |
| EthanolConcentration | 0.635 | 0.000 | 0.412 | 0.023 | 0.475 | 0.013 |
| FaceDetection | 0.614 | 0.009 | 0.638 | 0.008 | 0.616 | 0.008 |
| FingerMovements | 0.520 | 0.000 | 0.535 | 0.013 | 0.497 | 0.032 |
| HandMovementDirection | 0.270 | 0.000 | 0.493 | 0.031 | 0.380 | 0.027 |
| Handwriting | 0.371 | 0.000 | 0.585 | 0.004 | 0.510 | 0.006 |
| Heartbeat | 0.756 | 0.000 | 0.740 | 0.011 | 0.762 | 0.011 |
| JapaneseVowels | 0.922 | 0.000 | 0.966 | 0.002 | 0.987 | 0.004 |
| LSST | 0.382 | 0.039 | 0.639 | 0.003 | 0.652 | 0.004 |
| Libras | 0.900 | 0.000 | 0.906 | 0.004 | 0.922 | 0.009 |
| MotorImagery | 0.580 | 0.000 | 0.572 | 0.012 | 0.545 | 0.047 |
| NATOPS | 0.644 | 0.000 | 0.884 | 0.008 | 0.926 | 0.013 |
| PEMS-SF | 0.873 | 0.000 | 0.826 | 0.012 | 0.829 | 0.018 |
| PenDigits | 0.937 | 0.003 | 0.983 | 0.001 | 0.965 | 0.002 |
| Phoneme | 0.178 | 0.010 | 0.276 | 0.002 | 0.294 | 0.003 |
| RacketSports | 0.888 | 0.000 | 0.910 | 0.008 | 0.879 | 0.016 |
| SelfRegulationSCP1 | 0.761 | 0.000 | 0.849 | 0.005 | 0.917 | 0.006 |
| SelfRegulationSCP2 | 0.511 | 0.000 | 0.546 | 0.017 | 0.509 | 0.008 |
| SpokenArabicDigits | 0.964 | 0.003 | 0.997 | 0.001 | 0.994 | 0.002 |
| StandWalkJump | 0.600 | 0.000 | 0.524 | 0.023 | 0.358 | 0.041 |
| UWaveGestureLibrary | 0.909 | 0.000 | 0.937 | 0.004 | 0.936 | 0.006 |
| FD001 | 0.938 | 0.007 | 0.936 | 0.016 | 0.950 | 0.012 |
| FD002 | 0.888 | 0.006 | 0.931 | 0.004 | 0.913 | 0.012 |
| FD003 | 0.936 | 0.008 | 0.946 | 0.019 | 0.932 | 0.015 |
| FD004 | 0.824 | 0.008 | 0.879 | 0.008 | 0.877 | 0.012 |
| Mafaulda | 0.762 | 0.012 | 0.922 | 0.008 | 0.900 | 0.009 |

## A.3 NUMBER OF FEATURES SELECTION

In Figure 7 we can see the evolution of the accuracy with the increasing number of features for the L1L2 method, as measured in our randomly selected development subset of datasets. We see that a common pattern is that the accuracy initially increases, and reaches a plateau after a number of features (different for each dataset). Another pattern is the oscillation of accuracy around an initial value. The selection of 500 as the default number of features is based on this graph, and is a reasonable number which balances accuracy and method speed, as well as ensures that most development datasets are close to or in the plateau region.

Table 2: Detailed accuracy metrics of LightWaveS-L1, L2 and L1L2-1500 variants

| Dataset | L1 | | L2 | | L1L2-1500 | |
|---|---|---|---|---|---|---|
| | Mean | Std | Mean | Std | Mean | Std |
| ArticularyWordRecognition | 0.997 | 0.000 | 0.980 | 0.000 | 0.993 | 0.000 |
| AtrialFibrillation | 0.267 | 0.000 | 0.067 | 0.000 | 0.200 | 0.000 |
| BasicMotions | 1.000 | 0.000 | 1.000 | 0.000 | 1.000 | 0.000 |
| CharacterTrajectories | 0.980 | 0.000 | 0.983 | 0.000 | 0.987 | 0.000 |
| Cricket | 0.972 | 0.000 | 0.944 | 0.000 | 0.986 | 0.000 |
| DuckDuckGeese | 0.400 | 0.000 | 0.360 | 0.000 | 0.420 | 0.000 |
| ERing | 0.967 | 0.000 | 0.922 | 0.000 | 0.974 | 0.000 |
| EigenWorms | 0.809 | 0.000 | 0.962 | 0.000 | 0.969 | 0.000 |
| Epilepsy | 0.986 | 0.000 | 0.978 | 0.000 | 0.993 | 0.000 |
| EthanolConcentration | 0.551 | 0.000 | 0.559 | 0.000 | 0.631 | 0.000 |
| FaceDetection | 0.622 | 0.008 | 0.504 | 0.008 | 0.618 | 0.007 |
| FingerMovements | 0.580 | 0.000 | 0.490 | 0.000 | 0.530 | 0.000 |
| HandMovementDirection | 0.270 | 0.000 | 0.243 | 0.000 | 0.297 | 0.000 |
| Handwriting | 0.329 | 0.000 | 0.279 | 0.000 | 0.388 | 0.000 |
| Heartbeat | 0.766 | 0.000 | 0.761 | 0.000 | 0.737 | 0.000 |
| JapaneseVowels | 0.932 | 0.000 | 0.935 | 0.000 | 0.965 | 0.000 |
| LSST | 0.402 | 0.035 | 0.343 | 0.004 | 0.425 | 0.014 |
| Libras | 0.872 | 0.000 | 0.794 | 0.000 | 0.928 | 0.000 |
| MotorImagery | 0.490 | 0.000 | 0.520 | 0.000 | 0.520 | 0.000 |
| NATOPS | 0.644 | 0.000 | 0.633 | 0.000 | 0.622 | 0.000 |
| PEMS-SF | 0.896 | 0.000 | 0.861 | 0.000 | 0.879 | 0.000 |
| PenDigits | 0.943 | 0.003 | 0.922 | 0.002 | 0.958 | 0.001 |
| Phoneme | 0.179 | 0.008 | 0.143 | 0.018 | 0.222 | 0.004 |
| RacketSports | 0.862 | 0.000 | 0.763 | 0.000 | 0.888 | 0.000 |
| SelfRegulationSCP1 | 0.744 | 0.000 | 0.696 | 0.000 | 0.761 | 0.000 |
| SelfRegulationSCP2 | 0.539 | 0.000 | 0.522 | 0.000 | 0.533 | 0.000 |
| SpokenArabicDigits | 0.967 | 0.003 | 0.966 | 0.002 | 0.980 | 0.002 |
| StandWalkJump | 0.400 | 0.000 | 0.600 | 0.000 | 0.533 | 0.000 |
| UWaveGestureLibrary | 0.900 | 0.000 | 0.738 | 0.000 | 0.931 | 0.000 |
| FD001 | 0.958 | 0.010 | 0.941 | 0.005 | 0.944 | 0.006 |
| FD002 | 0.877 | 0.007 | 0.907 | 0.004 | 0.927 | 0.003 |
| FD003 | 0.965 | 0.009 | 0.935 | 0.010 | 0.956 | 0.008 |
| FD004 | 0.796 | 0.008 | 0.867 | 0.006 | 0.876 | 0.006 |
| Mafaulda | 0.745 | 0.010 | 0.680 | 0.014 | 0.833 | 0.010 |

Table 3: Inference speedup of LightWaveS with fixed seed (0) compared to ROCKET

| Dataset | Inference Speedup | | | | | | | |
|---|---|---|---|---|---|---|---|---|
| | L1 | | L2 | | L1L2 | | L1L2 1500 | |
| | Mean | Std | Mean | Std | Mean | Std | Mean | Std |
| ArticularyWordRecognition | 33.3 | 4.4 | 23.8 | 2.5 | 24.5 | 3.0 | 14.4 | 1.4 |
| AtrialFibrillation | 36.3 | 3.9 | 19.1 | 1.7 | 19.0 | 1.5 | 11.7 | 0.8 |
| BasicMotions | 28.5 | 5.4 | 19.3 | 2.4 | 21.4 | 3.7 | 12.0 | 0.8 |
| CharacterTrajectories | 28.1 | 4.6 | 19.9 | 1.4 | 20.6 | 2.7 | 13.0 | 0.6 |
| Cricket | 59.1 | 5.6 | 34.0 | 1.0 | 41.9 | 2.1 | 20.0 | 0.6 |
| DuckDuckGeese | 257.1 | 17.6 | 180.0 | 5.2 | 179.1 | 9.6 | 84.1 | 2.7 |
| ERing | 23.3 | 3.1 | 16.3 | 2.1 | 17.4 | 3.0 | 11.2 | 1.1 |
| EigenWorms | 68.9 | 3.1 | 41.1 | 1.0 | 41.3 | 1.1 | 17.4 | 0.6 |
| Epilepsy | 29.1 | 2.8 | 19.6 | 1.1 | 26.0 | 3.0 | 12.7 | 0.7 |
| EthanolConcentration | 44.8 | 2.9 | 28.3 | 1.2 | 37.3 | 1.6 | 16.1 | 0.7 |
| FaceDetection | 92.0 | 11.0 | 65.9 | 8.7 | 64.7 | 11.1 | 35.0 | 2.7 |
| FingerMovements | 39.2 | 5.4 | 27.2 | 4.4 | 28.1 | 3.9 | 16.3 | 1.6 |
| HandMovementDirection | 61.1 | 3.8 | 38.0 | 2.0 | 37.8 | 1.8 | 17.2 | 0.5 |
| Handwriting | 28.1 | 4.0 | 19.2 | 2.1 | 20.9 | 2.0 | 12.0 | 0.6 |
| Heartbeat | 67.3 | 5.2 | 42.3 | 2.5 | 44.6 | 1.8 | 17.6 | 0.7 |
| JapaneseVowels | 25.1 | 2.7 | 19.5 | 2.5 | 19.5 | 2.6 | 13.0 | 1.6 |
| LSST | 21.2 | 2.5 | 16.3 | 2.0 | 17.2 | 2.8 | 10.5 | 1.8 |
| Libras | 14.8 | 1.5 | 10.7 | 1.4 | 12.1 | 1.7 | 8.3 | 1.3 |
| MotorImagery | 79.0 | 2.0 | 46.6 | 1.0 | 47.7 | 1.0 | 18.0 | 0.4 |
| NATOPS | 37.1 | 4.4 | 25.3 | 3.6 | 29.2 | 4.2 | 16.9 | 1.7 |
| PEMS-SF | 283.1 | 36.2 | 207.3 | 19.8 | 205.9 | 21.8 | 93.9 | 3.4 |
| PenDigits | 9.4 | 2.1 | 8.3 | 1.1 | 8.9 | 1.4 | 6.8 | 0.9 |
| Phoneme | 47.4 | 5.3 | 30.1 | 2.1 | 31.7 | 1.7 | 14.7 | 0.4 |
| RacketSports | 19.8 | 2.4 | 14.1 | 1.8 | 15.0 | 1.9 | 8.8 | 1.0 |
| SelfRegulationSCP1 | 55.6 | 3.0 | 32.5 | 1.5 | 35.4 | 1.0 | 15.3 | 0.3 |
| SelfRegulationSCP2 | 59.7 | 2.7 | 35.6 | 1.0 | 37.6 | 1.0 | 15.1 | 0.4 |
| SpokenArabicDigits | 37.4 | 5.4 | 25.9 | 3.0 | 28.4 | 3.3 | 14.9 | 0.9 |
| StandWalkJump | 54.7 | 1.5 | 32.9 | 0.7 | 33.9 | 0.8 | 15.3 | 0.4 |
| UWaveGestureLibrary | 36.6 | 3.3 | 21.8 | 0.9 | 26.6 | 1.4 | 12.8 | 0.4 |
| FD001 | 23.4 | 2.1 | 20.2 | 2.9 | 20.1 | 2.8 | 13.8 | 1.8 |
| FD002 | 21.3 | 2.9 | 18.9 | 2.3 | 19.3 | 3.3 | 14.4 | 2.8 |
| FD003 | 30.2 | 4.0 | 23.7 | 4.0 | 25.6 | 3.4 | 16.0 | 1.9 |
| FD004 | 28.4 | 4.1 | 24.9 | 3.2 | 24.4 | 2.8 | 14.7 | 2.4 |
| Mafaulda | 55.2 | 8.1 | 36.4 | 4.6 | 37.6 | 6.1 | 16.2 | 1.4 |

Table 4: Inference speedup of LightWaveS with fixed seed (0) compared to MINIROCKET

| Dataset | Inference Speedup | | | | | | | |
|---|---|---|---|---|---|---|---|---|
| | L1 | | L2 | | L1L2 | | L1L2 1500 | |
| | Mean | Std | Mean | Std | Mean | Std | Mean | Std |
| ArticularyWordRecognition | 5.9 | 0.8 | 4.2 | 0.5 | 4.4 | 0.6 | 2.6 | 0.3 |
| AtrialFibrillation | 2.8 | 0.3 | 1.4 | 0.1 | 1.4 | 0.1 | 0.9 | 0.0 |
| BasicMotions | 6.5 | 1.3 | 4.4 | 0.6 | 4.9 | 0.9 | 2.7 | 0.2 |
| CharacterTrajectories | 4.5 | 0.8 | 3.2 | 0.3 | 3.3 | 0.5 | 2.1 | 0.1 |
| Cricket | 2.3 | 0.2 | 1.3 | 0.1 | 1.6 | 0.1 | 0.8 | 0.0 |
| DuckDuckGeese | 232.9 | 15.6 | 163.1 | 5.2 | 162.2 | 8.1 | 76.2 | 2.3 |
| ERing | 6.7 | 1.0 | 4.7 | 0.6 | 5.0 | 0.8 | 3.2 | 0.3 |
| EigenWorms | 1.6 | 0.1 | 0.9 | 0.0 | 0.9 | 0.0 | 0.4 | 0.0 |
| Epilepsy | 4.1 | 0.4 | 2.7 | 0.2 | 3.6 | 0.4 | 1.8 | 0.1 |
| EthanolConcentration | 1.5 | 0.1 | 0.9 | 0.0 | 1.3 | 0.0 | 0.5 | 0.0 |
| FaceDetection | 71.1 | 8.5 | 51.0 | 6.6 | 50.0 | 8.5 | 27.0 | 2.1 |
| FingerMovements | 19.9 | 2.7 | 13.8 | 2.3 | 14.3 | 2.0 | 8.3 | 0.8 |
| HandMovementDirection | 4.9 | 0.3 | 3.1 | 0.2 | 3.0 | 0.2 | 1.4 | 0.1 |
| Handwriting | 4.7 | 0.7 | 3.2 | 0.4 | 3.5 | 0.4 | 2.0 | 0.1 |
| Heartbeat | 13.5 | 1.1 | 8.5 | 0.5 | 8.9 | 0.4 | 3.5 | 0.1 |
| JapaneseVowels | 12.6 | 1.3 | 9.8 | 1.2 | 9.8 | 1.3 | 6.5 | 0.8 |
| LSST | 8.9 | 1.2 | 6.9 | 0.9 | 7.3 | 1.3 | 4.4 | 0.8 |
| Libras | 6.3 | 0.7 | 4.5 | 0.6 | 5.1 | 0.7 | 3.5 | 0.6 |
| MotorImagery | 4.7 | 0.1 | 2.8 | 0.1 | 2.8 | 0.1 | 1.1 | 0.0 |
| NATOPS | 17.6 | 2.1 | 12.1 | 1.7 | 13.9 | 2.1 | 8.0 | 0.8 |
| PEMS-SF | 258.2 | 33.3 | 189.0 | 17.6 | 187.7 | 19.8 | 85.7 | 3.0 |
| PenDigits | 7.9 | 1.9 | 7.0 | 1.1 | 7.5 | 1.4 | 5.7 | 0.9 |
| Phoneme | 7.4 | 0.8 | 4.7 | 0.4 | 5.0 | 0.3 | 2.3 | 0.1 |
| RacketSports | 9.6 | 1.2 | 6.8 | 0.9 | 7.2 | 0.9 | 4.3 | 0.5 |
| SelfRegulationSCP1 | 2.8 | 0.2 | 1.6 | 0.1 | 1.8 | 0.1 | 0.8 | 0.0 |
| SelfRegulationSCP2 | 2.5 | 0.1 | 1.5 | 0.1 | 1.6 | 0.0 | 0.6 | 0.0 |
| SpokenArabicDigits | 10.7 | 1.6 | 7.4 | 0.8 | 8.1 | 1.0 | 4.3 | 0.3 |
| StandWalkJump | 1.6 | 0.1 | 1.0 | 0.1 | 1.0 | 0.1 | 0.5 | 0.0 |
| UWaveGestureLibrary | 3.8 | 0.4 | 2.3 | 0.2 | 2.8 | 0.2 | 1.3 | 0.1 |
| FD001 | 14.5 | 1.5 | 12.5 | 1.9 | 12.5 | 1.7 | 8.5 | 1.1 |
| FD002 | 15.2 | 2.1 | 13.5 | 1.6 | 13.8 | 2.3 | 10.3 | 1.9 |
| FD003 | 17.4 | 2.4 | 13.6 | 2.3 | 14.7 | 1.9 | 9.2 | 1.1 |
| FD004 | 20.8 | 3.1 | 18.2 | 2.3 | 17.9 | 2.1 | 10.8 | 1.7 |
| mafaulda | 2.6 | 0.3 | 1.7 | 0.1 | 1.8 | 0.2 | 0.8 | 0.0 |

Table 5: Number of original dataset channels used by LightWaveS

| Dataset | Channels | Channels retained by method | | | | | | | |
|---|---|---|---|---|---|---|---|---|---|
| | | L1 | | L2 | | L1L2 | | L1L2 1500 | |
| | | Mean | Std | Mean | Std | Mean | Std | Mean | Std |
| AWR | 9 | 7 | 0 | 8 | 0 | 7 | 0 | 7 | 0 |
| AtrialFib | 2 | 2 | 0 | 2 | 0 | 2 | 0 | 2 | 0 |
| BasicMotions | 6 | 5 | 0 | 5 | 0 | 5 | 0 | 5 | 0 |
| ChTrajectories | 3 | 3 | 0 | 3 | 0 | 3 | 0 | 3 | 0 |
| Cricket | 6 | 6 | 0 | 4 | 0 | 6 | 0 | 6 | 0 |
| DuckGeese | 1345 | 102 | 0 | 124 | 0 | 98 | 0 | 174 | 0 |
| ERing | 4 | 4 | 0 | 4 | 0 | 4 | 0 | 4 | 0 |
| EigenWorms | 6 | 6 | 0 | 6 | 0 | 6 | 0 | 6 | 0 |
| Epilepsy | 3 | 3 | 0 | 3 | 0 | 2 | 0 | 3 | 0 |
| EthanolC | 3 | 3 | 0 | 3 | 0 | 3 | 0 | 3 | 0 |
| FaceDetection | 144 | 106.05 | 6.92 | 134.38 | 2.48 | 117.78 | 6.89 | 142.87 | 1.42 |
| FingerMov | 28 | 28 | 0 | 28 | 0 | 28 | 0 | 28 | 0 |
| HMD | 10 | 10 | 0 | 10 | 0 | 10 | 0 | 10 | 0 |
| Handwriting | 3 | 3 | 0 | 3 | 0 | 3 | 0 | 3 | 0 |
| Heartbeat | 61 | 46 | 0 | 51 | 0 | 42 | 0 | 58 | 0 |
| JapVowels | 12 | 6 | 0 | 10 | 0 | 4 | 0 | 10 | 0 |
| LSST | 6 | 4.9 | 0.84 | 4.83 | 0.69 | 4.5 | 0.97 | 5.7 | 0.46 |
| Libras | 2 | 2 | 0 | 2 | 0 | 2 | 0 | 2 | 0 |
| MotorImagery | 64 | 64 | 0 | 64 | 0 | 64 | 0 | 64 | 0 |
| NATOPS | 24 | 6 | 0 | 6 | 0 | 6 | 0 | 6 | 0 |
| PEMS-SF | 963 | 113 | 0 | 209 | 0 | 132 | 0 | 300 | 0 |
| PenDigits | 2 | 2 | 0 | 2 | 0 | 2 | 0 | 2 | 0 |
| Phoneme | 11 | 11 | 0 | 11 | 0 | 11 | 0 | 11 | 0 |
| RacketSports | 6 | 6 | 0 | 6 | 0 | 6 | 0 | 6 | 0 |
| SelfRegSCP1 | 6 | 6 | 0 | 6 | 0 | 6 | 0 | 6 | 0 |
| SelfRegSCP2 | 7 | 7 | 0 | 7 | 0 | 7 | 0 | 7 | 0 |
| SpArabicDigits | 13 | 8.47 | 0.57 | 8.83 | 0.38 | 8 | 0.52 | 9.27 | 0.45 |
| SWJump | 4 | 4 | 0 | 4 | 0 | 4 | 0 | 4 | 0 |
| UWaveGL | 3 | 3 | 0 | 3 | 0 | 3 | 0 | 3 | 0 |
| FD001 | 26 | 11.38 | 2.11 | 19.23 | 1.87 | 13.53 | 1.79 | 20.23 | 1.74 |
| FD002 | 26 | 21.63 | 0.49 | 20.2 | 0.76 | 21.23 | 0.5 | 22.93 | 0.83 |
| FD003 | 26 | 22.72 | 1.65 | 23.1 | 1.52 | 23.57 | 1.5 | 25.93 | 0.25 |
| FD004 | 26 | 19.73 | 0.52 | 25.07 | 0.94 | 20.33 | 0.96 | 25.97 | 0.18 |
| Mafaulda | 8 | 8 | 0 | 8 | 0 | 8 | 0 | 8 | 0 |

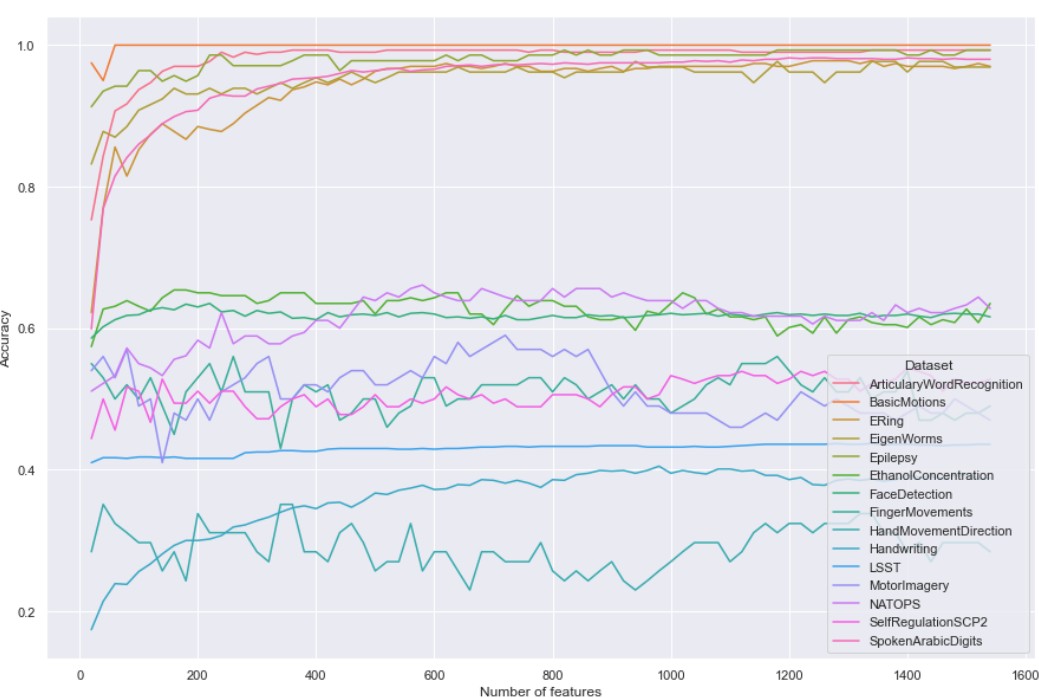

Figure 7: Evolution of accuracy with increasing number of features for L1L2 method

