# OpenReview forum: "Taking ROCKET on an efficiency mission: A distributed solution for fast and accurate multivariate time series classification"
_ICLR.cc/2022/Conference — ICLR 2022 Submitted_

### Official Review · Reviewer_Vj84 · 2021-10-20

**Correctness:** 2
**Technical Novelty And Significance:** 3
**Empirical Novelty And Significance:** 3
**Recommendation:** 5
**Confidence:** 4

**Main Review:**

The authors present a multivariate time series classification (MTSC) pipeline that is an improvement over the (MINI)ROCKET algorithm. The authors use wavelets in a clever way that requires only 2.5% of ROCKETs features. The authors also report averaged performances, which is the right practice and generates robust results.

I would say that the main weakness is that nothing is said about hyper-parameter searches. Thus, all the results are greatly dependent on whichever values were chosen. Hence, it is not possible to assess the generality of the presented results.

**Summary Of The Paper:**

The authors present the LightWaveS pipeline, whose training is comparable to that of (MINI)ROCKET and an inference time 8x to 30x times faster.

**Summary Of The Review:**

The authors present an optimized pipeline with good results, whose generality is not possible to assess.

---

> ### Author Response · Authors · 2021-11-23
> **Response to review comments**
>
> Dear reviewer,
>
> Thank you very much for taking the time to go through our manuscript and for your review.
>
> We purposefully did not do exhaustive hyperparameter search in order to avoid overfitting the solution to the tested datasets. The options that we choose are either guided by theory and intuition (e.g. the scattering format), are similar to the (MINI)ROCKET publications for more fair comparison (e.g. max dilations) or can be substituted with equivalent methods (e.g. ANOVA feature selection). The 3 features (percentage of positive values, long stretch over 0 and max) have been well tested in (MINI)ROCKET and MULTIROCKET publications, and the minimum is a simple feature added as an additional descriptor of the coefficients after the application of the modulus (abs value).
>
> The default number of features (500) has been selected according to our development dataset. We added a section in the appendix that shows that for most datasets, the accuracy either increases steadily with more features until it reaches a plateau, or it oscillates around an initial value. Thus, 500 was a good balance between accuracy and speed for most datasets.
>
> In general, given the speed of our method, we would expect a practitioner to do a gridsearch with these various options, adapted to their specific problem/dataset.

---

### Official Review · Reviewer_DUKD · 2021-10-29

**Correctness:** 2
**Technical Novelty And Significance:** 2
**Empirical Novelty And Significance:** 2
**Recommendation:** 3
**Confidence:** 4

**Main Review:**

Strengths:

1. Distributed training and inference is a timely and important problem for time-series tasks
2. Real-world datasets are used and the code is available
3. The method is simple and seems effective

Weakness

1. The solution is mostly a nice engineering effort but is lacking in terms of methodological results
2. The results are not impressive: there is a loss in accuracy

Details:

1. The solution seems a combination of two existing methods. There are claims of the generality of this methodology but no proof of that. How does it apply to other kinds of features? how well it works. The "random" kernels of ROCKET help to apply the exact same framework, but I am not sure the same methodology would help combine features from catch22 for example or other approaches.

2. How much is the benefit of your approach vs. using the Dask framework (or other parallelization frameworks) for MINIROCKET for example and perform inference by distributing model and fingering class per time series using a similar amount of resources. There are way simpler approaches to increase throughput once the model is built considering the significant training overhead your method is having.

3. There is a significant loss in accuracy. Even though it's not statistically significant, we can observe a significant loss. It's not significant because here you evaluate such a small number of datasets so there is no enough evidence to show significant better or worse results either way. but going from 2nd best to 8th best or 9th best indicates some significant loss. (CD lines cover the entire area...)

4. There is growing literature for DNNs once the model is trained to keep only a small subset of parameters/connections to significantly compress the models. Similar methodologies could be applied here as a lot simpler solutions (as post-processing steps once the models are built). Even though you make this connection with the work of Bruna and Mallat, there is a growing literature these days that is not mentioned or compared against that is critical for understanding if your path is the best/more meaningful than these more "modern" approaches.

See ideas from "deep compression" literature:

 Han, Song, Huizi Mao, and William J. Dally. "Deep compression: Compressing deep neural networks with pruning, trained quantization and huffman coding." arXiv preprint arXiv:1510.00149 (2015).

**Summary Of The Paper:**

The paper proposes LightWaveS, a distributed methodology for multivariate time-series classification based on wavelet scatting transformation. The method utilizes a small percentage of the initial set of features while achieving comparable accuracy performance but with significant speedup. The method is evaluated on several publicly available multivariate time-series datasets.

**Summary Of The Review:**

The method shows good speedup, however, there is a significant loss in accuracy too. DNN literature for compressing models is missing but seems relevant for this problem. The solution is not clear how it generalizes to other problems as claimed. No proof/experimentation is provided at least.

---

> ### Author Response · Authors · 2021-11-23
> **Response to review comments**
>
> Dear reviewer,
>
> Thank you very much for taking the time to go through our manuscript and for your review.
>
> 1. The intuition behind the generality (and orthogonality) claim is based on the relevant literature, e.g.  [1]. The scattering coefficients can be still viewed as time series and described with a large variety of features. Since feature selection is inherent to our method, we expect the most valuable of those for the classification task to be selected. The specific performance would depend on the problem and the selected feature types, but we view this flexibility as an advantage of our method, since the significant speed allows for quick experimentation with various features based on the practitioner’s experience and demands.
>
> [1] Tan, Chang Wei, et al. "MultiRocket: Effective summary statistics for convolutional outputs in time series classification." arXiv preprint arXiv:2102.00457 (2021).
>
> 2.	We assume that the computational resources are abundant and readily available during training (e.g. a cloud environment) and scarce during inference in less than ideal environments (e.g. an edge device in an industrial environment). In this case, it would be more complex to distribute the inference and the communication overhead would be possibly quite significant. This is why we consider a single edge device during inference.
>
> 3.	It is true that the accuracy of ROCKET is hard to match in all datasets, but our main objective was the speedup of the method, which is quite significant and disproportional to the loss. We also wanted to show that for the achieved training and inference time, apart from ROCKET, the solution stands well among recent deep learning solutions.
>
> 4.  This is indeed an interesting view of the problem. In this context our solution can be considered as “growing” from the ground up, as opposed to pruning/compression. Some additional important factors are the emphasis on the training speed, as well as the computational (memory as well as execution) impracticality of building all the paths in the first place, especially with datasets with more channels, or with more base wavelets/dilations. We believe that such an exploration of this problem may be better suited for future work.

---

> > ### Comment · Reviewer_DUKD · 2021-11-29
> > **Thanks for replying to our comments**
> >
> > I believe you are focusing on a very specific scenario (train with an abundance of resources but the inference is constrained resources), which is common and that is what creates issues. There are so many ways in which you can speed up inference, from hardware acceleration to parallelism to feature engineering tricks to dimensionality reduction to feature selection to statistical tests, etc. You chose one particular methodology, which is solid, but it's not convincing at what principle it was made. Why not a simple feature selection then? 2-3 other reviewers pointed out such approaches. I think focusing/comparing with such solutions will make the paper stronger.
> >
> > Also is ROCKET really the way to go? If yes, great. But if not, shouldn't we be thinking of solutions to generalize across different classifiers let's say? You are focusing on one specific solution but it does not mean in 2 years it will be the best. What are the abstractions across classification methods in that domain that we may want to accelerate? Focusing on that direction will be likely more impactful than just speeding up a standalone method.

---

### Official Review · Reviewer_PviB · 2021-11-05

**Correctness:** 3
**Technical Novelty And Significance:** 4
**Empirical Novelty And Significance:** 3
**Recommendation:** 6
**Confidence:** 4

**Main Review:**

Strengths:
1. The paper proposes an algorithm that deals with an important problem.
2. The experiments demonstrate the scalability and execution time benefits and show that the accuracy losses are not high.

Weaknesses:
1. Several aspects of the algorithm need further explanation:
     a. Why was the same set of kernels used for both levels of the two-level wavelet scattering? This should be justified and some comparison should be done with the case where different features are chosen at each level.
     b. Is algorithm 1 run at each level or at the top level? How are inputs distributed among the different nodes and how are outputs collected from the different nodes and merged? This needs explanation.
2. In figure 2 (part of experiments), what do the numbers 2000, 3000, 5000, and 10000 represent?

**Summary Of The Paper:**

This paper works to improve upon the ROCKET algorithm for multivariate time series classification. In particular, the proposed algorithm, LightWaveS, added wavelet scattering, convolutional kernels, and feature selection to make the algorithm much faster and more scalable while preserving close to the same accuracy. The algorithm can also be distributed and the experiments demonstrate good execution time reduction with additional computational nodes.

**Summary Of The Review:**

Overall, the paper does well at handling an important problem. However, there are some key details about the algorithm that need to be clarified and a key aspect of the experimental results need to be clarified.

---

> ### Author Response · Authors · 2021-11-23
> **Response to review comments**
>
> Dear reviewer,
>
> Thank you very much for taking the time to go through our manuscript and for your review.
>
> 1.	We modified section 3.2 to better explain the algorithm and justify our choices.
> 2.	The numbers refer to the number of features, after feature selection, for MINIROCKET. We moved this subsection to the Appendix, since it is useful but not crucial to the paper.

---

### Official Review · Reviewer_u4XS · 2021-11-05

**Correctness:** 3
**Technical Novelty And Significance:** 2
**Empirical Novelty And Significance:** 2
**Recommendation:** 5
**Confidence:** 3

**Main Review:**

The paper is largely well-written and provides a detailed analysis against several benchmarks. The idea of adding scattered wavelets to an existing model is interesting but it is not new as the authors have also cited related work in this domain.

The proposed method allows controlling the trade-off between complexity and accuracy when performing inference.

The presentation of results using the mean rank doesn't describe the accuracy of the model against the other models except for (Mini)Rocket. This is not an issue since (Mini)Rocket has the best mean rank, but it would have been very useful to see a full table of results; so that the reader can get a clear idea of the improvements or losses to accuracy/F1/etc they might expect.
Especially in some use cases, the trade-off between recall and precision will be very important. For example in an engine failure prediction, recall could have a much higher weight compared to precision.

While the wavelet scattering is well described, the motivation behind using it, except improving the speed, is not very clear. And it is not also clear if with reduced features or reduce accuracy (some of) the existing models could also provide the same level of accuracy.

A more intuitive explanation of the structure of the data that this method works on would have been very helpful. Real-world time-series data could be noisy, dynamic with deterministic or non-deterministic changes into its distribution and it would have been good to explain for what type of datasets and in what of the applications the proposed model could be useful.


**Summary Of The Paper:**

The paper describes a model for feature extraction and classifying multivariate time-series data. It proposes using scattered wavelet and multiple kernel analysis to extract features from multivariate time-series data. The model can run in a distributed manner and on multiple nodes. It uses a feature selection mechanism which also improves the interpretability of the method.

The work is built on the existing ROCKET and Mini-ROCKET models and instead of the kernel function used in ROCKET, it proposes using Wavelet Scattering.

The key contribution of the paper is modifying the ROCKET model and adding Wavelet Scattering and allowing a flexible number of kernels/features that can be extracted from each segment of the time-series data. This has contributed to improving the speed and scalability of the model compared to ROCKET. The work has also been extensively evaluated on the UEA benchmark datasets and two other separate datasets (one of these includes 5 sub-sets).

**Summary Of The Review:**

The paper is largely well-written and it is clear that the authors have put a lot of effort to implement and evaluate their work and benchmarking it again existing methods and have also tried it on new datasets.

The presentation of the results in terms of providing detailed Precision/Recall/F1/AUC would have been very helpful. Comparing the speed/complexity and/or performance of the other methods at the same level of accuracy would have provided a more balanced comparison.

Figure 6 is hard to read and comprehend. It seems the mean difference is shown in Figure 6(b); but it is not clear what for example a -0.3 difference means. Does that mean 30% lower accuracy?

Overall, the idea of wavelet scattering as a feature extraction is interesting. However, as the authors have mentioned "LightWaveS can act as an initial fast channel filtering method that precedes another deep learning solution"; the latter could have been a good starting point to start the work and build a processing pipeline.

In terms of code and reusability, a set of python code have been provided but it would have been very helpful if dependencies or a read me or a notebook code accompanied the code to make it more readable/re-usable.

---

> ### Author Response · Authors · 2021-11-23
> **Response to review comments**
>
> Dear reviewer,
>
> Thank you very much for taking the time to go through our manuscript and for your review.
>
> We have added detailed accuracy and F1 metrics in the appendix for our solution. Due to time constraints, we will add more detailed precision/recall metrics at a repository after the completion of the process. The same applies for the provided code, it was supplied as a reproducibility proof, and a more structured version with dependencies will be uploaded to a repository.
>
> The motivation behind developing and using wavelet scattering resulted from the insight of the wavelet connection with the (MINI)ROCKET solution and the related literature which suggests that more levels of coefficients are more descriptive, together with the aim for faster training and inference. Among the compared solutions, Catch22 and WEASEL+MUSE already have performed some kind of feature selection, and for the more complex DL ones, this would require a more sophisticated and adapted approach.
>
> We could perform a similar sophisticated feature selection on (MINI)ROCKET, but there is a qualitative difference, since we would select best features from the generated random feature space. Our solution exhausts the wavelet convolutions for the first layers and selects the second based on intuition and experiments (see also answer to reviewer zJvV), so it is not stochastic, and allows for the channel reduction and insight in the problem. Finally, since each ROCKET feature requires at least 1 convolution in the best case scenario, the speedup would still be present.
> Although we aim for a generalized method, we agree with the observation that a further exploration of the characteristics of the datasets whose accuracy benefits/decreases with the method is interesting, and maybe better suited for future work.
> In figure 6, the difference is indeed in terms of accuracy, so -0.3 corresponds to -30% accuracy.

---

### Official Review · Reviewer_ooqH · 2021-11-07

**Correctness:** 3
**Technical Novelty And Significance:** 2
**Empirical Novelty And Significance:** 2
**Recommendation:** 6
**Confidence:** 4

**Main Review:**

The paper provides promising directions for improving the inference time which is critical for edge devices; however, there are a couple of points that needs to be addressed first.

1) A major source of confusion in Algorithm1 is that non of the functions GetDataSlice(), GenerateKernels(), WaveletScatteringFeatures(), TopFeatureSelection(), GatherTopFeatures(), FeatureTransform(), and MRMRFeatureSelection() are defined and the descriptions within the manuscript are very high-level. It is recommended to provide more details about the functionality of these subroutines.

2) During the inference, every evaluation of the same dataset may take slightly different runtime, as it is usually always the case in practice. Hence, in order to show the improvement in inference speedup, as in Figure 6, authors need to repeat the inference test several times and report the mean and standard deviation (SD) of the inference speedup instead of just a single number. Hence, the simulations for Figure 6 needs to be repeated and both the average and SD should be reported. If the standard deviation is very small, Figure 6 can be modified to only contain the mean values; however, in either case, the result should be based on multiple tests and not just a single inference test.

**Summary Of The Paper:**

This paper tries to improve the inference time of deep learning models by proposing a distributed feature selection process based on wavelet scattering transformations of the time series. In particular, authors proposes a method called LightWaveS for a fast, distributed transformation of multivariate time series based on convolutional kernels, wavelet scattering and feature selection, for lightweight and accurate classification with linear classifiers. The end goal is to improve the inference time while maintaining the performance.

**Summary Of The Review:**

See the comments provided above.

---

> ### Author Response · Authors · 2021-11-23
> **Response to review comments**
>
> Dear reviewer,
>
> Thank you very much for taking the time to go through our manuscript and for your review.
>
> 1.	Based on your comment and other reviews we decided to remove the algorithm and clear up the explanation in the text.
> 2.	We have indeed run the inference (and all the experiments) multiple times. We have added detailed metrics with standard deviation in the appendix.

---

### Official Review · Reviewer_uVpJ · 2021-11-07

**Correctness:** 2
**Technical Novelty And Significance:** 2
**Empirical Novelty And Significance:** 2
**Recommendation:** 3
**Confidence:** 3

**Details Of Ethics Concerns:**

N.A.

**Main Review:**

Strengths :

LightWaveS offers comparable accuracy to the original ROCKET model, with training time similar to (MINI)ROCKET and inference time 8x-30x faster on an edge device.

Weaknesses :

1. The authors claim that they utilize a wavelet scattering transformation in the model as the main contribution. The wavelet is realized by using a dilation operation in convolutional kernels. There is only one sentence: “Intuitively, we can relate these ... wavelet ...” to illustrate the relationship between them, which is not convincing. The author should justify this important claim and provide theoretical explanations if any.

2. The author claim that the proposed LightWaveS is much faster than (MINI)ROCKET. The reason is very simple: (MINI)ROCKET uses the default number of features (20 and 10 thousand), and LightWaveS only uses 500 features. Obviously, the computational cost of the LightWaveS should be significantly less than (MINI)ROCKET. However, the authors only conduct feature selection on their method, what if feature selection is also conducted on (MINI)ROCKET?

3. The authors should conduct ablation studies on the features generated with wavelet scattering and the feature selection, and show which part contributes to the better accuracy/efficiency tradeoff more?

4. The figures are difficult to understand, e.g., some notations are not explained, “Count” in Fig.4. what are numbers mean? (0.05, 0.05-0.1)


**Summary Of The Paper:**

This paper presents a multivariate time series classification (MTSC) model, namely LightWaveS, that extends another MTSC framework ROCKET. In addition to inheriting the successful aspects of the ROCKET model family (accuracy and training time), it significantly improves inference time. To be specific, LightWaveS adds wavelet scattering, multi-node distribution, and "smart feature selection" to address the drawbacks of the ROCKET. Experiments on five machinery-related datasets show the efficacy of LightWaveS.

**Summary Of The Review:**

While the results are promising compared to existing ROCKET models, the proposed solution is not well explained and the experimental results are not fully justified.

---

> ### Author Response · Authors · 2021-11-23
> **Response to review comments**
>
> Dear reviewer,
>
> Thank you very much for taking the time to go through our manuscript and for your review.
>
> 1.	We tried to support this claim more clearly.
> 2.	(This answer is similar as the one to reviewer u4XS) We could indeed perform some sophisticated feature selection on (MINI)ROCKET and keep fewer features, however our approach is different in two aspects: First, we try to remove randomness and exhaust the search space in wavelets/dilation for the first layer and based on intuition/experiments for layer 2. This allows for the advantages such as the channel reduction and insight in the problem. However, in (MINI)ROCKET we would just select from the randomly generated feature space. Secondly, each ROCKET feature (with which we utilize the same kernel application mechanism) is based on at least 1 convolution, so for the same number of features the speedup would still exist, especially on datasets with larger number of channels, where the reduction of our solution is large.
> 3.	A type of this ablation is the L1 and L2 variants, which utilize the feature selection but are from different levels of the scattering. Regarding the feature selection, it would be computationally impractical to train a model with all the generated features, so we rely on the intuition and experimentation for the second level selection.
> 4.	We tried to clear up the Figure 4. “Count” referred to the number of datasets where we accomplish accuracy within the specified range as (MINI)ROCKET (e.g. 0.05 means within 5% points of accuracy)

---

> > ### Comment · Reviewer_uVpJ · 2021-11-29
> > **Thank you for the responses**
> >
> > The authors gave some explanations, but I don't think they are convincing in their current form.

---

### Official Review · Reviewer_zJvV · 2021-11-07

**Correctness:** 2
**Technical Novelty And Significance:** 2
**Empirical Novelty And Significance:** 2
**Recommendation:** 3
**Confidence:** 4

**Main Review:**

## Strengths:
The paper studies an important research direction.

## Weaknesses:
The presentation of the work is meandering and unfocused, some typos are present which negatively impact reading.

Multivariate time series classification is a very well studied topic. The proposed method, LightWaveS, is a relatively minor extension of the ROCKET method. The overall contribution of this work may be incremental. Moreover many design choices seem rather arbitrary and lack either intuition or rigor (see i-iii). E.g. what allows the authors to only consider the paths in bold in Figure 1 and ignore all the others? It is clear that this reduces computational complexity but it is not described what are the modeling issues related with this arbitrary choice.

**Major issues:**

**i-** the distributed implementation, while being counted as a contribution, to me seems rather straightforward and is not described in much detail. If the distributed implementation is not straightforward I believe it is worth to precisely describe it in the appendix, which at this stage is not present, and highlight the specific and unique, to this work, design choices the authors made.

**ii-** the connection with the scattering transform is not clear. The overall LightWaveS architecture is NOT a scattering network, nor an approximation in any way. The computational “trick” suggested by the authors (i.e. considering only the paths in bold in Figure 1) oversimplifies the proposed model. I believe it would be important to spend more words and perhaps perform an experimental evaluation (if no theory can be used) on this crucial aspect.

**iii-** What is the intuition behind the use of ANOVA as a general feature selection criterion? Why has L1 regularization or any other learnt subset of features not been compared? The choice of using ANOVA seems rather arbitrary and does not help the reader in gaining any deeper intuition on the design choice.

**iv-** Both features’ interpretability and incorporation of domain knowledge is explicitly cited in the abstract, I do not find experiments supporting this claim in the main text, these are briefly mentioned in the “Discussions and Conclusions” section. Please update the abstract or add a more in depth discussion (validated through proper experiments).


**Minor:**

What do authors precisely mean by: “there is efficient and minimal communication between the worker nodes and the central coordinator.”?

Figure 1 should be improved in quality, moreover its font should be corrected to be consistent with the main text.

The L1 and L2 versions of the algorithm should be considered as ablated models of the L1L2 model rather than fully fledged models.

**Experimental results:**
- Give some intuitions on why the L2 model is worse than the L1 one (in all the experiments).
- Missing appendix with extra details on datasets and “implementation” (e.g. distributed implementation and Ridge classifier).
- The reader would expect the experiment in Figure 2.b repeated for LightWaveS to test the capability of features selection of ANOVA.
- Add standard deviation on accuracy results.

**Typos and notation:**
- Bad notation in section 3.1.1.: “max” for the exponent used to sample the max length of dilation, and “b” is used both in the scattering transform and in Equation (1).
- Section 3.2: Be consistent with common literature on scattering transform: use subscripts (e.g. \lambda_1 in place of \lambda 1).
- In section 3.1.2, the following quotes:  ”child”.
- Algorithm 1 does not add much to the exposition and is not very clear (missing quantities such as L and typo on x in place of \times in the definition of the input \mathbf{X}).
- Part of the introduction is better suited for the related work section (too details which are hard to grasp for the reader a this time).
- Improve the writing of section 4.5: the last part is hard to read and understand, while the first part is trivial.


Here are some references which worth mentioning are:

[1] (NIPS 2017) Random Projection Filter Bank for Time Series Data

[2] (ICLR 2020) Classification-Based Anomaly Detection for General Data

**Summary Of The Paper:**

This paper presents a distributed algorithm to solve Multivariate time series classification (MTSC). The proposed solution LightWaveS is based on the ROCKET algorithm. The advancements over ROCKET are: feature selection and distributed implementation. The authors show these advancements allow LightWaveS to outspeed both ROCKET and MINIROCKET (a faster variant of MINIROCKET) in inference.

**Summary Of The Review:**

The presentation of the work is meandering and unfocused, some typos are present which negatively impact reading.
Multivariate time series classification is a very well studied topic. The proposed method, LightWaveS, is a relatively minor extension of the ROCKET method. The overall contribution of this work may be incremental. Moreover many design choices seem rather arbitrary and lack either intuition or rigor (see i-iii). For these reasons I believe the manuscript does not meet ICLR's standards and should be rejected.

---

> ### Author Response · Authors · 2021-11-23
> **Response to review comments**
>
> Dear reviewer,
>
> Thank you very much for taking the time to go through our manuscript and for your review.
>
> i. The distributed implementation is indeed straightforward in terms of the data method, so we do not present it as a contribution per se, but rather as an integral part in the context of the entire solution, which allows the quick exploration of the feature space. We try to describe the hierarchical feature selection, which is the part that is not intuitive.
> ii. Since we were driven by the speedup of the method first and foremost, we tried to limit the spread of the layer 2 to the bare minimum (so one path). The intuition was to apply the same wavelet to the L1 output, in order to look for the same pattern in lower frequency, and it gave satisfactory results. Prompted by your review, we experimented with various heuristic methods to select the L2 kernel (such as the one that gave the best feature overall or per feature category) and larger spreads (2 - 4 L2 kernels), but those methods gave similar (and slightly worse) results to our first approach, so the added complexity is not justified. We added this explanation in the paper.
> iii. ANOVA was selected as a feature selection method due to it being a fast filter method to rank the features. The main driver for the selection is the speed of the method, and any other filter feature selection algorithm could be substituted, which gives the framework flexibility. More complex methods such as getting the useful features from models trained with regularization or subset creation would significantly affect the training time.
> iv. We agree with your observation. The domain knowledge and interpretability refer to the potential of the solution, which is the selection of suitable wavelets and the easy interpretation of the convolution result and meaning of the features, respectively. We acknowledge that this is not demonstrated by experiments in this work, so we modified the abstract.
>
> We address the “Minor” comments in a suitable manner.
>
> (This comment is similar to the one to reviewer PviB) Although in some cases the L2 model is better, it would be indeed interesting to explore which datasets benefit or are hurt a lot when selecting features from different levels, but we believe it would be better suited for future work.
>
> We added the appendix with detailed metrics.

---

> > ### Comment · Reviewer_zJvV · 2021-11-29
> > **Thank you for the responses**
> >
> > I thank the authors for their answers, nonetheless I do not believe point i and iii have been fully addressed.
> >
> > Overall, some more modifications and the use of precise statements will certainly improve the quality of the paper, which, as of now, I believe does not meet the ICLR standard.

---

### Author Response · Authors · 2021-11-23
**General comments after reviews**

We would like to thank all reviewers for their valuable input. We tried to address the points raised as best we can. In addition, prompted by the reviews and with the inclusion of the detailed metrics in the appendix, we have also updated our reported speedup (in the Abstract and main text body) to accurately reflect that of the datasets for which we achieve comparable accuracy with ROCKET (better or within 5%).
During additional experimentation, we decided to limit randomness even more during feature selection, so we also updated the supplementary material and figures based on the new metrics. All the conclusions are still valid and the variants now show slightly better performance.

---

### Decision · Program_Chairs · 2022-01-20

**Decision:**

Reject

**Comment:**

This paper proposes an algorithm called LightWaveS to improve the ROCKET (and mini-ROCKET) algorithm for multivariate time series classification, by using wavelet scattering instead of the kernel function. More than the usual number of reviewers were invited to provide independent reviews on the paper.

A concern was raised regarding the lack of hyperparameter search in the paper. The authors responded that this was intentional to avoid overfitting the solution to the tested datasets. This response is not convincing. Note that other important reasons to vary the hyperparameter values (as commonly adopted by ML researchers) are to study the sensitivity of the proposed method to hyperparameter settings and to perform more holistic performance comparison with other methods.

Other concerns on both novelty and significance have also been raised.

Although 2 of the 7 reviews show a weak support for acceptance, other reviewers have pointed out legitimate concerns that make this paper not ready for publication in ICLR in its current form. We appreciate the authors for clarifying some points in their responses and discussions and even including further results, but addressing all the concerns raised really needs a more substantial revision of the paper. We hope the comments and suggestions made by us can help the authors prepare a revised version that will be more ready for publication.